# Fetal Bovine Serum Supplementation Enhances Functional Consistency of IGRA Results in Bovine Tuberculosis Diagnostics [note 1]

**DOI:** 10.3390/ani15172580

**Published:** 2025-09-02

**Authors:** Jae-Kyo Jeong, Mi-Na Lim, Ki-Joo Kim

**Affiliations:** Jeonbuk State Institute of Veterinary Service and Research, Southern Branch, Namwon-si 55725, Republic of Korea; edenfarms@korea.kr (M.-N.L.); kkj6060@korea.kr (K.-J.K.)

**Keywords:** cell-mediated immunity test, T-cell immune response, diagnostic sensitivity, mycobacterium bovis, culture supplements

## Abstract

Early and accurate detection of bovine tuberculosis is essential to protect both animal health and the livestock industry. A widely used blood test called the interferon-gamma release assay (IGRA) can detect this disease by measuring the immune response of white blood cells. However, this test often becomes less reliable when there is a delay between blood collection and laboratory analysis. In this study, we investigated whether adding a nutrient-rich supplement called fetal bovine serum (FBS) could help preserve immune cell function during such delays. Blood samples from cattle were tested with and without this supplement after being stored for three days. We found that samples treated with FBS showed stronger immune responses and a higher detection rate compared to those without FBS. This suggests that FBS can help stabilize the test results and reduce the chances of missing infected animals. Our findings are important because they show that a simple and low-cost addition to the testing procedure can make bovine tuberculosis diagnosis more reliable, especially in real-world settings where immediate testing may not always be possible. This approach could ultimately improve disease control and benefit both farmers and public health.

## 1. Introduction

Bovine tuberculosis (bTB), caused by *Mycobacterium bovis*, is a chronic zoonotic disease that poses significant risks to animal health, public safety, and international trade. Human infection can occur through direct contact with infected cattle or the consumption of unpasteurized animal products. For these reasons, bTB is designated as a notifiable disease by the World Organization for Animal Health (the WOAH, formerly OIE) and remains a major target of eradication programs worldwide [1]. In the Republic of Korea, 4138 confirmed cases were reported between 2022 and 2024, with Jeollabuk-do—one of the nation’s major Hanwoo cattle production areas—accounting for 590 cases despite ongoing control measures [2]. This persistent incidence underscores the need for sensitive, reliable diagnostics that can maintain accuracy during the interval between on-farm sample collection and laboratory processing, including transport and routine delays.

The tuberculin skin test (TST), or single intradermal comparative cervical tuberculin (SICCT) test, is the most widely used screening method for live cattle and is recognized by the WOAH as a standard diagnostic tool [3,4]. However, its sensitivity and specificity vary with environmental, host, and reagent factors [5,6]. The interferon-gamma release assay (IGRA) has emerged as a complementary or alternative method, offering higher sensitivity and specificity, quantifiable results, and the ability to detect *M. bovis*-specific T-cell responses using antigens such as ESAT-6 and CFP-10 [7,8]. Because IGRA depends on the viability and functional integrity of live T cells, diagnostic accuracy is vulnerable to pre-analytical factors such as delays in testing, suboptimal storage temperatures, or inadequate culture conditions [9,10], which can lead to false-negative results.

Fetal bovine serum (FBS) is a widely used cell culture supplement that provides nutrients and stabilizing factors to support cell survival. While its benefits are well established in vitro, its potential to preserve T-cell responsiveness for IGRA under delayed testing conditions has not been systematically evaluated. A previous study has shown that FBS can modulate cytokine production in immune-related cells, potentially increasing baseline responses even without a specific antigenic stimulation [11]; yet, few studies have investigated whether supplementation can reduce functional signal loss or improve IGRA diagnostic outcomes, and most have assessed FBS only as part of general culture media rather than as an independent variable during sample storage.

This study aimed to evaluate whether FBS supplementation can preserve immune cell function, enhance diagnostic sensitivity, and maintain reproducibility of IGRA results under delayed processing conditions via the parallel assessment of antigen-specific and mitogen-induced responses.

## 2. Materials and Methods

### 2.1. Sample Collection

Whole-blood samples were obtained from Hanwoo cattle over 12 months of age from two farms in Jeollabuk-do, Republic of Korea. Both herds underwent complete testing under the national tuberculosis and brucellosis control program following the detection of a confirmed or presumptive bovine tuberculosis case during pre-movement screening. The first farm, which provided 91 samples, had multiple IGRA-positive animals at initial screening, and follow-up testing confirmed that 51 out of 91 animals (56.0%) were positive; this herd was designated IGRA-positive for the study. From this herd, a subset of 13 animals was selected solely on the basis of sufficient residual blood volume after routine IGRA testing, without regarding optical density values or response patterns, to minimize selection bias. The second farm, sampled on a different date and in a separate area, provided 22 samples, all negative in both initial and follow-up testing; all animals from this herd were included in the analysis. In addition to the delayed storage comparison between FBS-unsupplemented (FBS X) and FBS-supplemented (FBS O) conditions, an additional Day 0 paired comparison was performed in this group to assess whether FBS alone influenced immediate IGRA results.

Blood was aseptically collected into sodium heparin-coated tubes. For the FBS-supplemented condition, 100 µL of heat-inactivated fetal bovine serum (Gibco, Thermo Fisher Scientific, Waltham, MA, USA) was added to 900 µL of whole blood (final 10% *v*/*v*). Samples for delayed storage were aliquoted into deep-well plates and kept at 4 °C for 72 h before IGRA processing, while Day 0 samples were processed immediately.

### 2.2. IGRA Procedure

For each sample condition (Day 0, FBS X, FBS O), 1 mL of whole blood was stimulated with either an *M. bovis* antigen cocktail (ESAT-6, CFP-10, TB10.4), phosphate-buffered saline (negative control), or phytohemagglutinin (positive mitogen control). All reagents were from a commercial IGRA kit (Piron-gamma TB ELISA, Pigenomics, Daejeon, Republic of Korea; APQA license No. 236-001). After adding 50 µL of the stimulant, the tubes were incubated at 37 °C for 20 h, centrifuged at 500× *g* for 10 min, and plasma was harvested. Interferon-gamma levels were measured according to the kit protocol using a microplate reader (SpectraMax M2, Molecular Devices, San Jose, CA, USA) at 450 nm with a 620 nm reference. An optical density (OD) ≥ 0.1 was considered positive.

### 2.3. MTT Assay

T-cell metabolic activity was assessed in the same 13-animal subset used for delayed-condition IGRA analysis. Peripheral blood mononuclear cells were isolated using Leucosep tubes (Greiner Bio-One, Kremsmünster, Austria; and Sigma-Aldrich, St. Louis, MO, USA), resuspended in PBS, and plated in duplicate at 100 µL per well in a 96-well plate. The MTT reagent (Sigma-Aldrich, St. Louis, MO, USA) was added to achieve 0.5 mg/mL, and plates were incubated for 3 h at 37 °C. Formazan crystals were dissolved in 100 µL of dimethyl sulfoxide per well, and absorbance was recorded at 570 nm.

### 2.4. Statistical Analysis

Statistical analyses were conducted in R v4.5.1 (R Foundation for Statistical Computing, Vienna, Austria). OD distributions were evaluated with the Shapiro–Wilk test; all values deviated from normality (*p* < 0.05), so Wilcoxon signed-rank tests were used for paired comparisons between Day 0, unsupplemented delayed storage (FBS X), and FBS-supplemented delayed storage (FBS O). IGRA positivity was defined as *M. bovis*-stimulated OD ≥ 0.1. Dual recovery was defined as an increase from Day 0 in both *M. bovis*– and mitogen-stimulated ODs for the same animal. Positivity and dual recovery frequencies were compared using chi-square or exact binomial tests, and the IGRA classification agreement was assessed with pairwise Cohen’s κ (95% CI).

Spearman’s rank correlation (ρ) was used to assess the following: (1) Day 0 mitogen-stimulated OD vs. Day 0 MTT OD, and (2) post-storage ΔOD (*M. bovis* and mitogen) vs. MTT OD under FBS X and FBS O. Significance was set at α = 0.01. Diagnostic sensitivity, specificity, PPV, and NPV were calculated using Day 0 results as the reference standard, as no independent gold standard was available.

## 3. Results

In this study, the term *bovis* refers to a recombinant *Mycobacterium bovis*–specific antigen cocktail containing ESAT-6, CFP-10, and TB10.4, which is provided as a standardized component of the commercial IGRA kit [12,13]. All samples were stimulated with the bovis antigen cocktail and mitogen according to the manufacturer’s instructions. Among 91 Hanwoo cattle over 12 months of age sampled from an IGRA-positive farm in Jeollabuk-do, 50 animals (54.9%) tested IGRA-positive on Day 0, and 41 animals (45.1%) tested negative (Appendix A).

After 72 h of storage at 4 °C, optical density (OD) values were consistently higher under the FBS-supplemented condition (FBS O) compared to the non-supplemented condition (FBS X). In the Bovis-stimulated condition (Figure 1A), most data points lay above the diagonal *y* = *x* line, and the difference between FBS X and FBS O was statistically significant (*p* = 1.91 × 10^−6^, Wilcoxon signed-rank test). A similar pattern was observed under mitogen stimulation (Figure 1B), with an identical *p*-value. The dotted lines in both scatter plots indicate the IGRA positivity threshold (OD = 0.1).

For consistent interpretation, three aspects were evaluated: (1) the proportion of Day 0-positive animals that remained positive after storage; (2) the proportion of Day 0-negative animals that showed increased IFN-γ responses under FBS supplementation; and (3) “Dual Recovery”, defined as simultaneous increases from Day 0 in both Bovis- and mitogen-stimulated OD values. Compared to Day 0, FBS supplementation improved sensitivity from 26.0% (FBS X) to 52.0% (FBS O) while specificity remained at 100% (Appendix A). Among the 50 Day 0-positive animals, dual recovery occurred in 42% under FBS X and 70% under FBS O (*p* = 0.0056, McNemar’s test). In the Day 0-negative group (*n* = 41), rates were 26.8% and 41.5%, respectively (*p* = 0.19). OD values for both stimuli declined after storage without FBS but were partially restored with supplementation (Table 1). In Day 0-positive animals, 26 remained bovis-positive under FBS O versus 13 under FBS X, with dual recovery experienced in 88.5% and 53.8% of these, respectively (*p* = 0.000044 vs. 0.11) (Table 2).

Within the 91-head cohort, a subset of 13 animals was further analyzed with paired IGRA–MTT data to examine classification agreement and its relationship to T-cell metabolic activity. Agreement analysis of IGRA classifications showed that κ improved from fair (Day 0 vs. FBS X, κ = 0.234) to moderate (Day 0 vs. FBS O, κ = 0.478) and was moderate-to-substantial between FBS X and FBS O (κ = 0.593) (Table 3), indicating that FBS supplementation enhanced reproducibility under delayed testing conditions.

MTT assays in these 13 animals showed that cold storage reduced T-cell metabolic activity, but this decline was partially mitigated in the FBS O group (*p* < 0.01, Wilcoxon signed-rank; Figure 2). Correlations between MTT OD and mitogen-stimulated IFN-γ OD at baseline were weak (Table 4), and post-storage ΔOD values under either condition showed minimal association with metabolic activity (|ρ| < 0.14).

Finally, in a separate IGRA-negative herd (*n* = 22), FBS supplementation generally increased OD values for both stimuli after storage, with greater increases for the mitogen stimulation (Table 5). Correlations between MTT OD and ΔOD for both stimulation types were again weak (|ρ| < 0.14; Table 6), suggesting that, in initially negative animals, FBS primarily preserved general T-cell responsiveness rather than inducing specific *M. bovis* reactivity.

Taken together, the quantitative improvements, higher agreement, and greater recovery rates observed across experimental conditions indicate that FBS can mitigate the functional decline of T cells during delayed IGRA processing, providing a basis for the mechanistic interpretation that follows.

## 4. Discussion

IGRA’s dependence on live, functional T cells makes it vulnerable to delays and suboptimal storage, which can lead to false negatives [14,15,16]. This study evaluated whether supplementation with 10% fetal bovine serum (FBS)—a standard cell culture additive—could preserve T-cell functionality during delayed IGRA processing. Using paired samples from the same animals under Day 0, FBS X, and FBS O conditions, the results showed that FBS supplementation improved both quantitative responses (higher IFN-γ optical densities) and qualitative outcomes (greater dual recovery and higher κ agreement). Comparable patterns of concurrent changes in both antigen- and mitogen-induced IFN-γ responses have been reported in earlier IGRA studies, albeit without targeted interventions to promote recovery [17,18,19]. For example, Waters et al. observed that holding blood samples for 8–24 h without supplementation resulted in concurrent declines in both responses in nearly all cases [20]. These parallels suggest that FBS supplementation can raise diagnostic reproducibility to levels already recognized as operationally relevant in epidemiological applications.

FBS is a widely used cell culture supplement known to support cell viability and functional capacity across diverse cell types [21,22,23,24,25]. In this study, supplementation enhanced IGRA responsiveness to both Bovis-specific and mitogen stimulation (Figure 1A,B), indicating partial restoration of T-cell function beyond simple survival. Unlike prior work by Gerace et al. [26], which adjusted the FBS concentration only at the point of antigen stimulation, the present design maintained 10% FBS throughout the 72 h storage period and directly compared paired IGRA samples with and without supplementation. Although previous studies have shown that FBS can influence reproducibility and modulate T-cell phenotypes in vitro [27,28,29], this represents the first explicit assessment of its role in preserving T-cell function and diagnostic reproducibility under delayed IGRA testing conditions.

Additional evidence for the cell-preserving role of FBS was derived from the MTT assay, which showed that storage without FBS was associated with a marked decline in metabolic activity, whereas supplementation mitigated this decline. The weak correlation between metabolic activity and IFN-γ production suggests that preservation of T-cell functionality may involve mechanisms beyond viability, such as stabilization of membrane integrity, antioxidant activity, or maintenance of intracellular signaling pathways [30,31,32]. For example, Lindroos et al. demonstrated that serum source influences membrane integrity and gene expression in cultured stem cells [30], while Mun et al. reported the antioxidant and stage-specific ROS-modulating effects of FBS in early pig embryo development [31].

Beyond improving overall IGRA positivity, FBS supplementation enhanced both the consistency and quality of immune responses. A greater proportion of animals in the FBS O group showed “Dual Recovery”—defined as simultaneous increases from Day 0 in both bovis- and mitogen-induced responses—which reflects a qualitative stabilization of T-cell function alongside quantitative increases in IFN-γ output. As the mitogen response serves as an intrinsic quality control in IGRA, the concurrent preservation of both specific and general immune pathways further supports this assay’s reliability under delayed conditions.

In contrast, the effect of FBS was minimal in animals that were initially IGRA-negative on Day 0. While some exhibited strong mitogen responses after storage with FBS, bovis-specific reactivity remained largely unchanged. Consistent with previous observations, FBS can enhance baseline cytokine production and modulate the basal inflammatory state of immune-related cells, indicating that serum availability itself influences resting immune responsiveness [33,34]. These findings suggest that baseline immune status and prior antigen exposure may determine the restorative potential of FBS.

To rule out the possibility that the observed enhancement was due to artificial stimulation by FBS itself, an additional Day 0 experiment was performed. Although responses in FBS-supplemented samples were slightly elevated, none exceeded the diagnostic cutoff. This confirms the fact that FBS does not induce false positives and can be safely incorporated into IGRA workflows without compromising specificity.

The 10% FBS concentration used in this study was based on standard lymphocyte culture protocols [11]. While appropriate for proof-of-concept validation, broader applicability will require further investigation into dose–response relationships and optimization for routine field use. FBS is already widely used in veterinary diagnostic laboratories and vaccine production—often at concentrations between 2% and 10%—to support viral replication, buffer culture conditions, and promote cell survival. It is also a standard supplement in diverse research, biotechnology, and pharmaceutical applications [35,36,37], underscoring its practicality for incorporation into routine IGRA workflows.

This study has certain limitations. We tested only a single FBS concentration, with a fixed storage duration of 72 h, and a constant refrigerated temperature of 4 °C. A preliminary attempt at higher storage temperatures (37 °C) resulted in extensive hemolysis and unusable samples, confirming the temperature sensitivity of IGRA. Although these results were excluded from the main dataset, they highlight the importance of stringent pre-analytical controls. Future work should examine different storage durations, temperatures, and FBS concentrations. Additionally, high-resolution immunophenotyping and cytokine profiling could help clarify the cellular mechanisms underlying the stabilization effect.

Temperature control is particularly important. A recent study reported that IGRA performance is highly sensitive to ambient temperature at blood collection; both excessively low and high temperatures reduced IFN-γ responses, while optimal positivity occurred when maximum daily temperatures were between 6 and 16 °C or minimum temperatures were between 4 and 7 °C [38]. These findings further emphasize the need for precise temperature management from collection to laboratory processing to ensure diagnostic accuracy. Taken together, these results indicate that FBS supplementation offers a practical and effective strategy to maintain IGRA accuracy under real-world field conditions.

## 5. Conclusions

This study evaluated whether fetal bovine serum (FBS) supplementation can preserve the diagnostic performance of the interferon-gamma release assay (IGRA) for bovine tuberculosis under delayed testing conditions. Using paired samples, FBS supplementation significantly restored IFN-γ responses to both antigen-specific and mitogen stimuli, improving positivity retention and dual recovery in Day 0-positive animals. These results demonstrate that FBS not only supports cell survival but also stabilizes T-cell functionality, thereby maintaining the reliability and reproducibility of IGRA. Given that the study used field-collected samples, FBS supplementation represents a practical and effective approach to enhance IGRA accuracy in routine surveillance. Future research should optimize FBS concentration and storage parameters to facilitate broader application.

## Figures and Tables

**Figure 1 animals-15-02580-f001:**
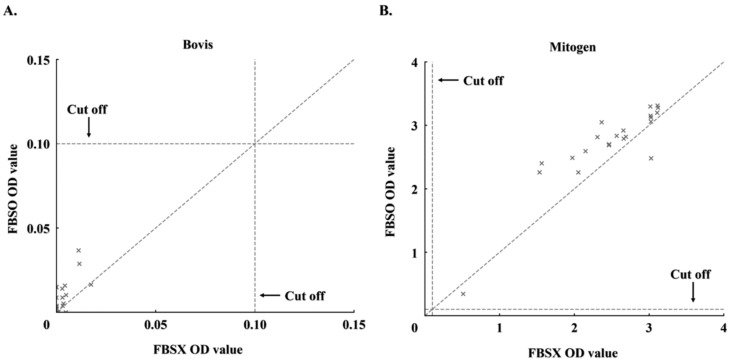
Scatter plots showing IFN-γ optical density (OD) values (*n* = 22) under FBS X (x-axis) and FBS O (y-axis) conditions. (**A**) *M. bovis*-stimulated samples; (**B**) mitogen-stimulated samples. The dashed line indicates the IGRA cutoff (OD = 0.1); the identity line (*y* = *x*) is shown for reference. Most points lie above the identity line, indicating increased IFN-γ responses with FBS. A statistical comparison was performed separately using the Wilcoxon signed-rank test (*p* < 0.01).

**Figure 2 animals-15-02580-f002:**
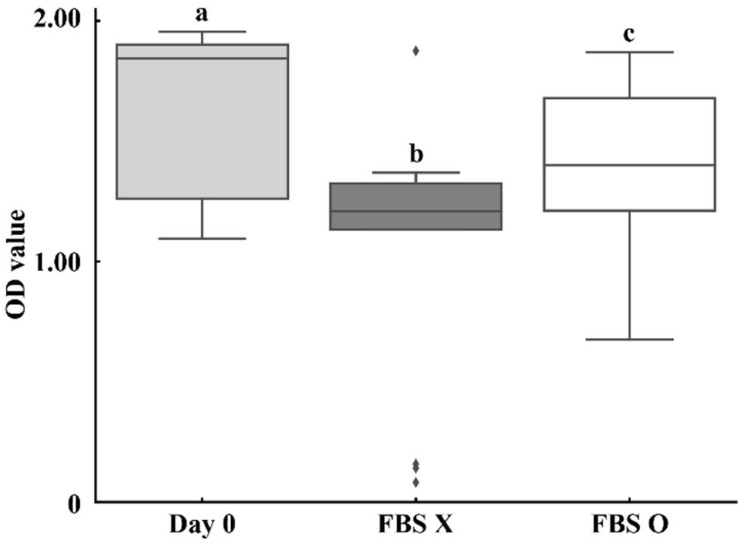
Optical density (OD) values from MTT cell viability assay under three conditions: Day 0 (fresh sample), FBS X (stored for 3 days without FBS), and FBS O (stored for 3 days with 10% FBS). Data are presented as medians with interquartile ranges in box plots (*n* = 13), with individual sample values indicated by rhombus symbols. FBS supplementation mitigated the reduction in cell viability during cold storage. Different letters (a, b, c) indicate statistically significant differences among groups (*p* < 0.01, one-way ANOVA with Tukey’s post hoc test).

**Table 1 animals-15-02580-t001:** Dual recovery after delayed storage with or without FBS, using baseline IGRA results.

Day 0 IGRA Result	Total Animals	Dual Recovery in FBS X	Dual Recovery in FBS O	Significance
Positive (*n* = 50)	50	21 (42.0%)	35 (70.0%)	FBS X: NS, FBS O: *p* < 0.01
Negative (*n* = 41)	41	11 (26.8%)	17 (41.5%)	FBS X: NS, FBS O: NS
Total	91	32 (35.2%)	52 (57.1%)	FBS X: NS, FBS O: *p* < 0.01

NS = not significant (*p* ≥ 0.01).

**Table 2 animals-15-02580-t002:** IGRA positivity and dual recovery in 50 initially IGRA-positive cattle.

Classification	No. of Animals	Positive by Bovis OD	Dual Recovery	Statistical Significance
Day 0 (baseline)	50	50 (100%)	—	—
FBS X (3-day w/o FBS)	50	13 (26%)	7 (53.8% of positives)	NS
FBS O (3-day with FBS)	50	26 (52%)	23 (88.5% of positives)	*p* < 0.01

FBS X = fetal bovine serum not supplemented; FBS O = fetal bovine serum supplemented; OD = optical density; NS = not significant (*p* ≥ 0.01).

**Table 3 animals-15-02580-t003:** Kappa coefficients for IGRA classification agreement.

Comparison	Kappa (κ)	Interpretation
Cohen’s kappa (Day 0 vs. FBS X)	0.234	Fair agreement
Cohen’s kappa (Day 0 vs. FBS O)	0.478	Moderate agreement
Cohen’s kappa (FBS X vs. FBS O)	0.593	Moderate agreement

**Table 4 animals-15-02580-t004:** Baseline (Day 0) Spearman correlation between lymphocyte viability (MTT OD) and mitogen-stimulated IFN-γ OD (*n* = 13).

Condition	Spearman’s ρ	*p*-Value
Day 0	0.109	0.712
FBS X	−0.039	0.899
FBS O	0.134	0.663

ρ = Spearman’s rank correlation coefficient; FBS = fetal bovine serum.

**Table 5 animals-15-02580-t005:** IFN-γ OD responses in 22 IGRA-negative cattle with or without FBS supplementation on Day 0.

Sample ID	Bovis FBS X	Bovis FBS O	Mitogen FBS X	Mitogen FBS O
1	0.0004	-	2.6891	2.8212
2	0.0174	0.0164	3.0240	3.1263
3	-	-	2.0524	2.2593
4	-	-	2.6553	2.9188
5	0.0047	-	2.6596	2.7924
6	0.0029	0.0140	2.4655	2.7038
7	-	0.0038	2.4600	2.6867
8	-	0.0023	3.1172	3.2793
9	-	0.0148	2.1481	2.5934
10	-	0.0150	1.5613	2.4027
11	0.0042	0.0159	3.0158	3.2982
12	-	0.0087	3.0264	2.4819
13	-	0.0038	3.0198	3.1529
14	0.0115	0.0288	3.1078	3.1986
15	0.0011	0.0003	1.5327	2.2602
16	0.0035	0.0052	2.3069	2.8167
17	0.0031	0.0039	2.3638	3.0506
18	0.0111	0.0367	3.1129	3.3138
19	-	0.0032	1.9733	2.4887
20	-	0.0023	3.0247	3.0598
21	0.0030	0.0088	2.5658	2.8353
22	0.0049	0.0102	0.5112	0.3446

– = not detected (value below detection threshold); FBS X = fetal bovine serum not supplemented; FBS O = fetal bovine serum supplemented at 10%; OD = optical density;

**Table 6 animals-15-02580-t006:** Spearman correlations between lymphocyte viability (MTT OD) and IGRA ΔOD (post-storage—Day 0) for Bovis- and mitogen-stimulated responses under FBS X and FBS O (*n* = 13).

Comparison	Spearman ρ	*p*-Value
ΔOD (Bovis) vs. MTT (FBSX)	0.137	0.655
ΔOD (Bovis) vs. MTT (FBSO)	−0.06	0.845
ΔOD (Mitogen) vs. MTT (FBSX)	0.137	0.655
ΔOD (Mitogen) vs. MTT (FBSO)	0.11	0.721

## Data Availability

The original contributions presented in this study are included in the article and Appendix A. Further inquiries can be directed at the corresponding author.

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
