# Peer review of "Fetal Bovine Serum Supplementation Enhances Functional Consistency of IGRA Results in Bovine Tuberculosis Diagnostics†"

_animals, 2025, doi:10.3390/ani15172580_

Round 1

Reviewer 1 Report (Previous Reviewer 1)

Comments and Suggestions for Authors

The authors satisfactorily addressed the concerns raised during the manuscript review. As a minnor comment, Figure 2 could be included in the manuscript with higher resolution.

Author Response

Comment: The authors satisfactorily addressed the concerns raised during the manuscript review. As a minor comment, Figure 2 could be included in the manuscript with higher resolution.

Response: We sincerely appreciate the reviewer’s constructive suggestion. In response, Figure 2 has been regenerated at 300 dpi in TIFF format with enlarged font size to improve readability. The revised figure has been incorporated into the manuscript accordingly, and the reference to “Figure 2” has been highlighted in the text for clarity.

Reviewer 2 Report (Previous Reviewer 2)

Comments and Suggestions for Authors

I think the changes made made a positive contribution to the work.

WRITING AND FORMAT: Minor changes to scientific writing should be made... sentences cannot begin with "We found...". I suggest changes to "the authors found"

Reword the entire text where sentences appear in the first/third person.

Comments on the Quality of English Language

Need revision.

Author Response

Comments : I think the changes made made a positive contribution to the work.
WRITING AND FORMAT: Minor changes to scientific writing should be made... sentences cannot begin with "We found...". I suggest changes to "the authors found". Reword the entire text where sentences appear in the first/third person.

Response : We revised all sentences in the Discussion and Conclusion sections that previously contained first-person expressions (e.g., “we found,” “we maintained,” …). These were rephrased into objective, data-centered forms (e.g., “the results showed,” “10% FBS was maintained,” …). Revisions were made at lines 155, 167, 213, 227, 230, 256, and 286 to ensure a consistent scientific writing style.

This manuscript is a resubmission of an earlier submission. The following is a list of the peer review reports and author responses from that submission.

Round 1

Reviewer 1 Report

Comments and Suggestions for Authors

Line 69 y 70: It would be better to add more details about the paired comparison to finish this sentence. It was used for statistical analysis of the results?.

Line 71: subtitle is not in bold letter.

Line 72-74: Please, specify concentration and manufacturer for the antigen cocktail and mitogen. PPDb was also used for stimulation, add the same information about this reagent in this section.

Line 76: It would improve the manuscript if the authors add more information and citations or references to previous studies performed using this IFNg ELISA kit (cut-off value determination, etc.).

Line 86: The authors use Wilkoxon paired t test to compare values form Day 0 vs FBS 0 or FBS X. OD Data was analysed to confirm or reject normal distribution of the population? Please add information about this statistical test.

Line 90: It is not clear the meaning of this last sentence.

Line 96: Complete name of PPD should be added in the first mention of this reagent in the text. Moreover, it would be better to specify that the PPD is from M. bovis (PPDb).

Line 108: Mention the statistical test and p value in the figure legend.

Line 114-119: The authors performed lineal regression analysis to compare values of MTT with ELISA OD of FBS 0 and FBS X. As a suggestion, I think that this analysis will be more informative if the authors also perform lineal regression between MTT values and the Delta OD value (FBS 0 – FBS X) for each sample.

Line 124: Add the statistical test and p value in the figure legend. Also, it would improve the figure if the authors add lines and asterisks to highlight the groups with statistical differences.

Line 128: it would be helpful to begin this paragraph with more information about this sample collection (animal breed, age, location, etc.)

Line 133: “False-negative reduction”. In my opinion the authors should reconsider the name of this group. I assume that the ELISA kit has a defined protocol an a cut-off value determined, then samples under this cut-off at Day 0 should be considered as negative. The term false negative could be confused with positive animals defined by a gold standard method (M. bovis isolation, necropsy). This comment is in line with the previous suggestion to add more information for the ELISA kit (Line 76).  

Line 141: I consider that it would be more appropriate to replace “Bovis” by PPDb stimulated samples, or PPDb (same in table 2, line 144, line 175, etc. ).

Line 173: What were the selection criteria for these 22 negative animals?.

Line 195-203; This are the same animals described in line 173?. This paragraph describes the same results presented in line 176-188?.

Line 238-255: This paragraph reiterates findings already presented in the result section.

Line 261: Storage temperature should be considered as another variable (other options could be to storage samples at room temperature or 37°).

General comments

Sample collection: The manuscript would benefit from additional details regarding the herd/farm from which samples were collected, including TB prevalence and animal age. Furthermore, regarding the 22 negative animals discussed in the final results section: were these from the same sampling event, or a follow-up sampling of the same cohort?.

Results: The authors could have assessed the impact of FBS incubation on treated samples, as specified in the kit protocol (Day 0). A comparative analysis of PPDb (or mitogen)-stimulated values with and without FBS would have been valuable to determine whether FBS contributes to the increase in false-positive results.

Discussion: The results with the negative animals at Day 0 that turn positive for the test after the incubation period (with and without FBS) could be further discussed by considering studies in which samples were evaluated beyond the manufacturer's recommended storage period or that analyse false negative results for IGRA.

Reviewer 2 Report

Comments and Suggestions for Authors
  • INTRODUCTION: It lacks some background on bovine tuberculosis and its connection to immune status and IGRA as a diagnostic test. It should also include information on how other TB diagnostic tests work and the importance of IGRA.
  • M&M: The characterization of the collection location is missing... it only mentions "southern Jeollabuk-do"... it lacks information about the geographic location and a brief explanation of the characterization of cattle production in the area and the prevalence of TB and what routine tests are performed...
  • RESULTS and DISCUSSION: 

    There's a slight confusion about what constitutes results and discussion... because the discussion should include a comparison between the results of the work and others published previously... this comparison and the use of references are missing.

    Several sentences from the discussion need to be moved to the results and the references updated.

  • REFERENCES: see the format and update! The number of articles used is insufficient and does not support the work... an article must cite at least 30 to 35 articles...
Comments on the Quality of English Language

No comments.

Reviewer 3 Report

Comments and Suggestions for Authors

Dear authors,

Please find below the remarks, comments and suggestions resulting from the evaluation of your submitted manuscript. This report concerns the publication eligibility assessment of your study and its accompanying work as presented in your manuscript and aims in the detection, clarification and amendment of any flaws and/or omissions in order to improve its quality according to high scientific and publishing standards.

Line 42: I will have to ask you to rephrase “under field conditions”. It is rather unclear or ambiguous the intents meaning that you wish to communicate, since the  diagnostic test or procedure in question is presumably applied under laboratory conditions. I believe you were referring, most probably, to the conditions describing the routine procedure of sampling on farm (field) and the whole process and time intervening from collection, analysis and results delivery.

Line 47: I believe the fact of using the commonly used concentration of FBS, as alleged, as a reasonable pre-estimation is slightly related to the ability to enhance your study’s external validity. There are methods that can contribute in so, but in general your study's findings should be capable of being generalized to other populations, settings, and conditions, making the results more applicable and deducible to a reference population. In order to achieve so your study population should be drawn from a representative target population subgroup. The major impact on external validity can be accomplished by implementing a design allowing for the use of probability sampling and study’s reproducibility.

Line 51: As state above, the greater weight on the measures for assuring external validity for a study its is sampling design. Here, you are just mentioning a size of 91 animals, for which nothing is further elaborated on the way, methodology and reason that this size was selected. Are these animals from one farm or more? Were corresponding samples collected on the same day of slaughterhouse operation or not? You should describe in detail and report the procedure for sample selection, sample size determinations, any assumptions that were considered.

Lines 86-90:

You are reporting the use of a specific test, a non-parametric one specifically, without any prior mention or testing of the distribution of your outcome variable,-s. Was the normality assumption tested (provide results) for recorded parameters and how? Assumingly, they should not have been normally distributed. Additionally, Wilcoxon signed-rank test compared as stated ODs of INF-γ among “treatment” groups; were all possible two-way comparisons tested, namely Day0 vs FBS X, Day0 vs FBS O and FBS X vs FBS O. If this is the case, specify so.

I believe, however, there is a major flaw or omission, if you like, in this approach. You intend and should estimate not only differences between groups but also agreement, to measure and quantify inter-rater or more precisely inter-essay reliability to establish the degree of concordance and reproducibility  of results between the methods, that is between the FBS X and FBS O with the considered reference test that is Day0. However, there is a misconception of what is being compared and for what purpose. As very lucidly and vividly described by Ranganathan et al., 2017 (Common pitfalls in statistical analysis: Measures of

Agreement, Perspectives in Clinical Research | Volume 8 | Issue 4 | October-December 2017), analysis of pairs of measurements, either both categorical or both numeric, with each pair having been made on one individual may appear to be amenable to analysis by using methods for 2 × 2 tables (if the variable is categorical) or correlation (if numeric). In those methods, the two measurements on each individual relate to different variables, whereas in the “agreement” studies, the two measurements relate to the same variable. The basic concept to be understood here is that “agreement” quantifies the concordance between the two examiners or methods for each of the “pairs” of scores and not the similarity of the overall “success”” percentage between the examiners or methods (see examples in suggested article). Your analysis simply compares the proportion of positive results between the pre-defined groups. However, this is neither adequate nor complete in terms of reaching the desired result / answer as set in the study’s objectives. The fact that the compared frequencies differ or not merely suggests that proportion of positives results are not the same or are different, which does not necessarily implies that the result of a specific sample is the same or similar across the three different methods of testing applied. You should resort in the calculation of Cohen’s kappa to demonstrate inter‑essay agreement (between your groups) taking into account the expected agreement by chance, or more accurately Fleiss’ kappa, since this method is used when ratings by more than two observers (i.e. essays, tests etc.) are available for either binary or ordinal data.

This way your analysis will be complemented in a more accurate and complete way.

Line 93: Please specify the origin of these 20 cattle. Are they included from the initial sample size of 91, or a different sub-sample?

Line 105 (Figure 1): Scatter plots by no means compare, because they cannot. They are just graphically presenting or summarizing data in terms of a relation between two parameters. Please amend.

Line 110: These 13 cattle are part of the initial 91 or the previously mentioned group of the 20 IGRA-negatively classified cattle? Please clarify.

Line 112: You are reporting as  measures of central tendency, to describe and summarize your data, the average values of OD. Yet, as stated you used s non-parametric method (Wilcoxon signed-rank test), that compares median values). Thusly, it would be more informative and proper to report the medians and range of OD values.

Lines 114-119 and Table 1: The estimation of Pearson’s correlation coefficient has some prerequisites, in the form of distributional assumptions that have to be fulfilled. Most importantly, since the assumptions of independence within groups is valid, at least one of the two variables for which the analysis is conducted, must be normally distributed and with constant variance across the values of the other variable.

  • Which variable is normally distributed?
  • If so, why for this variable a parametric test instead of a non-parametric test (in this case statistical power is lost) was not applied / used?
  • Is Pearson’s correlation coefficient appropriate or you should have resorted to an approach free of any distributional assumption (e.g. Spearman’s r).
  • There is an issue of inconsistency in the statistical methodology used in terms of distributional criteria; if normality is not met then non-parametric tests should be used in all cases.

Table 1: What about Day0? Why wasn’t this group included in this analysis?

Line 123: No such data can be depicted in a box-plot graph. Box-plots do not report or depict means and SDs. The rectangular box is the middle 50% of the values (IQR), the horizontal, perpendicular line contained in the box is the median and the lines (or “whiskers”) extend up to 1.5 times the IQR length. This graph can show the distribution of values, especially under some kind of stratification, along with the presence of any outliers. Please amend accordingly.

Table 2: Interestingly, in this table your dataset is practically provided!

Table 3: You cannot rely solely on statistical difference (or not) to infer and support your hypotheses; as previously stated you also need agreement estimation and quantification (see previous comment). I can accept that recovery of less than the initial (Day 0) 50 positive results means loss of some valuable information, translated as you stated “false negatives” (by the way nothing has actually been provided for diagnostic performance of the essays used – actual Se & Sp) or a difference between FBS X and FBS O, means indirectly that the latter method can identify as “positives” more real positives than the former, yet you still need agreement analysis in order to cover all potential two-way interpretations and practically to demonstrate the suitability and reproducibility that the enriched medium that you propose offers compared to the golden standard (Day 0).

As a general and more relevant to the scope of your study comment is the fact that practically and diagnostic flexibility that can by provided by the suggested intervention is practically overlooked. As stated IGRA is considered a widely applied diagnostic test for bovine tuberculosis, offering higher Se and Sp and moreover served a reference standard for the serum supplementation you investigated. It would be very interesting and informative to provide the actual Se and Sp values for IGRA, and if this essay can be deemed as a golden standard for determining infection status to provide the achieved diagnostic performance of the other tested methods, ie. FBS O and FBS X.  I suggest extending your analyses to not only the suggested, alleged functional enhancement of IGRA results but to also to the quantification of both the agreement and performance of the stabilizing effect of FBS on T-cell function in IGRA compared to the widely applicable and standardized technique described. This way the assessment of its practical diagnostic applicability (line 56) will be fully accomplished.